# NECTIN4 (PVRL4) as Putative Therapeutic Target for a Specific Subtype of High Grade Serous Ovarian Cancer—An Integrative Multi-Omics Approach

**DOI:** 10.3390/cancers11050698

**Published:** 2019-05-20

**Authors:** Christine Bekos, Besnik Muqaku, Sabine Dekan, Reinhard Horvat, Stephan Polterauer, Christopher Gerner, Stefanie Aust, Dietmar Pils

**Affiliations:** 1Department of Obstetrics and Gynecology, Comprehensive Cancer Center (CCC), Medical University of Vienna, 1090 Vienna, Austria; christine.bekos@meduniwien.ac.at (C.B.); stephan.polterauer@meduniwien.ac.at (S.P.); stefanie.aust@meduniwien.ac.at (S.A.); 2Department of Analytical Chemistry, University of Vienna, 1090 Vienna, Austria; besnik.muqaku@univie.ac.at (B.M.); christopher.gerner@univie.ac.at (C.G.); 3Department of Pathology, Medical University of Vienna, 1090 Vienna, Austria; sabine.dekan@meduniwien.ac.at (S.D.); reinhard.horvat@meduniwien.ac.at (R.H.); 4Section for Clinical Biometrics, Center for Medical Statistics, Informatics, and Intelligent Systems (CeMSIIS), Medical University of Vienna, 1090 Vienna, Austria; 5Department of Surgery, Comprehensive Cancer Center (CCC), Medical University of Vienna, 1090 Vienna, Austria

**Keywords:** Nectin 4, NECTIN4, PVRL4, tumor spread, miliary or non-miliary, fallopian tube secretory epithelial cells (FTE), ovarian surface epithelial cells (OSE)

## Abstract

In high grade serous ovarian cancer patients with peritoneal involvement and unfavorable outcome would benefit from targeted therapies. The aim of this study was to find a druggable target against peritoneal metastasis. We constructed a planar—scale free small world—co-association gene expression network and searched for clusters with hub-genes associated to peritoneal spread. Protein expression and impact was validated via immunohistochemistry and correlations of deregulated pathways with comprehensive omics data were used for biological interpretation. A cluster up-regulated in miliary tumors with NECTIN4 as hub-gene was identified and impact on survival validated. High Nectin 4 protein expression was associated with unfavorable survival and (i) reduced expression of HLA genes (mainly MHC I); (ii) with reduced expression of genes from chromosome 22q11/12; (iii) higher BCAM in ascites and in a high-scoring expression cluster; (iv) higher Kallikrein gene and protein expressions; and (v) substantial immunologic differences; locally and systemically; e.g., reduced CD14 positive cells and reduction of different natural killer cell populations. Each three cell lines with high (miliary) or low NECTIN4 expression (non-miliary) were identified. An anti-Nectin 4 antibody with a linked antineoplastic drug–already under clinical investigation–could be a candidate for a targeted therapy in patients with extensive peritoneal involvement.

## 1. Introduction

High grade serous ovarian cancer (HGSOC) is a difficult to treat cancer entity, usually diagnosed at an advanced stage. The characteristic local peritoneal tumor spread is surgically limiting and followed by therapy resistant local (peritoneal) tumor recurrence as the predominant cause of cancer- associated death. Several molecular subclassification systems have been established [1,2,3,4,5] but did not lead to clinically relevant targeted therapies and over the last years little improvement in overall survival has been achieved.

Recently, evidence emerged that HGSOC can be divided into two groups according to different types of local tumor spread behavior, one with high involvement of the peritoneal cavity (typically presenting with numerous millet sized small implants on the peritoneal wall, called “miliary”) and one with low involvement of the peritoneal cavity (called “non-miliary”) (*cf*. Figure 1A) [6,7,8,9,10,11]. This spread behavior was linked to the putative origin of the cancer, fallopian tube secretory epithelial cells (FTE) or ovarian surface epithelial (OSE) cells [12,13,14]. Patients with miliary spread type and patients with tumors of putative tubal origin have an unfavorable survival outcome [6,8,12,15,16] and would urgently require new targeted therapies. It was also shown that patients with non-miliary tumor spread present with an increased adaptive immune reaction (e.g., CD8^+^ immune cell tumor infiltration) and immune-checkpoint expression (e.g., PD-L1 on tumor cells) [7], therefore might render good candidates for an additional immune therapy. Nevertheless, the link of high peritoneal, i.e., miliary, involvement with putative tubal origin and worse outcome [6,8,12] did not lead to plausible new therapy targets so far. As development of new therapeutics from scratch upon detection of new target candidates lasts for several years if not decades, the aim of this analyses was to identify targets for which putative therapeutics have been developed and tested in–at least–phase I clinical studies.

In the following study we aimed to searched for a druggable target for a potential treatment of patients with miliary HGSOC using own and publicly available RNA-sequencing data, to validate the impact of the target on overall survival with immunohistochemistry with an independent patient cohort and to biologically interpret the impact of this target on HGSOC using an innovative integrative multi-omics approach.

## 2. Results

### 2.1. Co-Association Gene Expression Clusters Associated with Tumor Spread Types

The aim of this analysis was to find a relevant over-expressed gene in patients with miliary tumor spread (associated with a significantly worse outcome) as defined in Auer et al. [6]. Therefore, our own RNA-sequencing data from tumors with known peritoneal tumor spread characteristics (miliary or non-miliary) were correlated with functional gene expression sub-clusters derived from a TCGA high grade serous ovarian cancer co-expression network. Clusters with up-regulated hub-genes already targeted by therapeutics in clinical studies were preferred, mainly to speed up potential clinical usage. As a first validation step the candidate sub-cluster was investigated via gene-signature analyses for its impact on overall survival in several other HGSOC cohorts with available gene-expression, clinicopathologic, and outcome data.

Whole transcriptome RNA-sequencing data from isolated tumor cells (i.e., enzymatically singularized and enriched for EpCAM protein abundance) of solid tissues and from ascites of patients with either miliary or non-miliary tumor spread were characterized according to three gene signatures: a 272 miliary-versus-non-miliary gene signature [6], a 211 ovarian-versus-tubal origin gene signature [12] (usually highly correlated with each other), and a 212 epithelial-mesenchymal (EM) gene signature [17]. Eight partly immortalized “normal” ovarian surface (four, three of them immortalized) and tubal secretory (four immortalized) epithelial cells were also analyzed accordingly. In Figure 1B results of these analyses are shown in triplots, clearly showing a discrimination in all three parameters of samples from patients with clinically evident miliary or non-miliary tumor spread (i.e., unambiguously classified intrasurgically [6]), most pronounced in ascites tumor cells. In solid tumor tissues no difference in the EM-status was revealed, similar to the non-immortalized OSE cells compared to the immortalized OSE cells (Figure 1B, left triplot). This probably is due to the growth condition as solid tumor tissues (or as epithelia on the surface of ovaries), which forces cells to a more epithelial status. In Figure 1A typical representations of both peritoneal spread types, miliary and non-miliary, are shown.

In Figure 2 the complete process and the results are shown, described briefly in the following sections. Whole transcriptome data of 303 HGSOC samples were normalized and used for building of a co-association network with about 4.3 million edges between the 20,501 genes with mutual information criterion as measure for the association (Figure 2A). Planar filtering of this network revealed a network of 47,581 edges connecting all genes in one large scale-free (degree distribution follows a power law) small world network (Figure 2B). This network was used for multiscale clustering analysis to get sub-networks (clusters) significantly more associated internally compared to the total network. Hub-genes of these clusters were also identified (Figure 2C). Finally, 653 hierarchically nested clusters were defined, with node numbers larger than ten and smaller than halve of the entire network (Figure 2D). Zero to 245 hub-genes were identified for each cluster. These 653 clusters were used for gene-set analyses with samples from ascites (A/S; *cf*. Figure 1B) and solid tissues (P/M; *cf*. Figure 1B) between patients with miliary and non-miliary tumor spread (Figure 2E; Appendix A). Finally, significant values of both comparisons were combined and corrected for multiple testing by the False Discovery Rate (FDR) approach. Interestingly, 149 out of 201 significant clusters below an FDR of 0.01 showed a matching direction of global deregulation in both comparisons (A/S and P/M), i.e., up or down in both, compared to only 400 out of all 653 clusters (enrichment *p*-value < 0.001). As for therapeutic targeting overexpressed genes are much easier druggable, the clusters up-regulated in miliary tumors were searched for hub-genes already used as targets in clinical studies in decreasing order of significance (Table 1).

Already the third cluster revealed an interesting candidate, i.e., cluster *c1_143* with 42 genes: Nectin 4 (*NECTIN4* or *PVRL4*) as only hub-gene in a network of nearly all up-regulated genes (31 out of 42, four down in both and seven either not expressed (*n* = 3) or differentially regulated in both comparisons) (Figure 2F). This cluster is a sub-cluster of the much larger second cluster in the table of significance-sorted deregulated clusters (i.e., cluster *c1_32* with 201 genes and six hub genes: CYHR1 (both up), EXOSC4 (up/down), SHARPIN (both up), NECTIN4 (both up), HSF1 (down/up), ZC3H3 (both down); Appendix A and links therein). Nectins and Nectin-like molecules (Necl) are families of immunoglobulin-like cellular adhesion molecules involved in Ca^2+^-independent cellular adhesion important for adherence and tight junctions [18,19]. To validate the impact of the cluster with Nectin 4 as hub-gene a gene signature value was calculated from the median expression of all genes up (in both comparisons) in miliary (*n* = 31) in this cluster minus the median expression of all genes down (in both comparisons) in miliary (*n* = 4) (Figure 2G and Appendix A).

The predictive value was dichotomized at the 25 percentile, taking into account our observation that approximately 75% of FIGO III/IV HGSOC patients present with miliary tumor spread and according to published data of frequencies of putative ovarian and tubal origin of HGSOCs [15,20]. This gene signature was applied to six publicly available gene expression data sets of ovarian cancer samples [4,21,22,23,24] meeting the following criteria: the clinicopathologic parameters grade, histology, FIGO stage, age at diagnosis, and residual tumor mass after debulking surgery had to be present; the first three parameters were used for selection of samples: Grade ≥ 2, serous histology, and FIGO stage ≥ III (in words: high grade late stage serous) the last three were used in multiple Cox-regression models to correct for these predictive clinicopathologic factors, together with the dichotomized Nectin 4 cluster gene signature value. A meta-analysis over these six independent data sets was significant for the dichotomized Nectin 4 cluster signature in an univariate setting (Figure 2H left; *p* = 0.003), and a trend in multiple Cox regression models together with clinicopathologic parameters FIGO stage, age, and residual tumor (Figure 2H right; *p* = 0.063, only one data set predicts favorable OS for Nectin 4 high samples in the multiple Cox-regression), showing an unfavorable outcome of patients with high Nectin 4 cluster expression.

### 2.2. HGSOC Cell Line Models

Forty-eight ovarian cancer cell lines [25] were also characterized with above mentioned gene signatures for tumor spread and origin and the NECTIN4 gene expression (Appendix A). Several high grade serous ovarian cancer cell lines with putative different spread and origin characteristics and appropriate Nectin 4 expressions (high in miliary of tubal origin, Caov-3, OVCAR-3, and OVKATE, and low in non-miliary of ovarian origin, OV-90, ES-2, and TYK-nu, with mean 36-fold higher Nectin 4 expression in the former) were identified, which could be used in further functional in vitro and in vivo (mouse) experiments.

### 2.3. Protein Expression of Nectin 4 (Assessed by Immunohistochemistry) and Impact on Overall Survival

Aim of this section was to validate the independent impact on overall survival of the candidate over-expressed hub-gene at the protein level (via immunohistochemistry staining of FFPE tissues) to strengthen the rational for using this gene product as target for a targeted therapy in selected patients with HGSOC.

Nectin 4 protein expression was assessed on 90 late stage (FIGO III/IV) HGSOC tissues by immunohistochemistry and scored according to percentage of positive tumor cells and staining intensities (H-score; 0, 1+, 2+, and 3+) (Figure 3 and Appendix A). The Cohen’s Kappa coefficient for the H-score between both raters was κ = 0.78 with ±1 as largest discrepancy. Samples with discordant H-scores were assessed together upon agreement. Percentage positive cells correlated with R = 0.91 and were averaged. Correlation of Nectin 4 protein expression to tumor spread types, miliary and non-miliary, was not possible as this information is not available from this cohort of patients. But gene expression analysis of all Nectin 4 cluster genes (*c1_143*, see Figure 2F) revealed clearly an up-regulation of the complete cluster in association to high Nectin 4 protein expression (Figure 4B,C, the latter showing the log_2_ fold-change enrichment barcode plot of the cluster). As there was no information available from literature concerning Nectin 4 impact on overall survival (OS) in HGSOC, we exploratively assessed the optimal cutoff by non-linear modeling of the Nectin 4 impact on OS by fractional polynomials Cox regression estimation [26] correcting for the known clinicopathologic factors age, FIGO stage, grade, and residual tumor mass after debulking surgery. In Figure 4A the corrected relative hazard against the percentage of Nectin 4 positive tumor cells is shown indicating a negative impact on OS in tumors with >50% and therefore this cutoff was used for outcome analyses. For all molecular biologic analyses (*cf.*
Section 2.4. and Figure 5) a trichotomized Nectin 4 protein expression score was used (0, negative; 1, ≤50%; and 2, >50%) to increase statistical power. Patient characteristics related to Nectin 4 expression are presented in Table 2. Incomplete surgical resection was significantly associated with a high Nectin 4 score (*p*-trend = 0.020; Table 2). In addition, a high Nectin 4 score was significantly associated with a higher frequency of neoadjuvant chemotherapy mode (overall 13.8%), indicating a higher tumor burden at diagnosis, but not with pre-operative CA-125 or ascites volume (Appendix A). All patients received a systemic platinum-based chemotherapy, either only adjuvantly of both, neo-adjuvantly and adjuvantly, the latter only if primary debulking surgery was not possible due to high tumor burden. The Nectin 4 score was not significantly associated to platinum-based chemotherapy resistance (defined as recurrent or progressive disease within six months after debulking surgery or neo-adjuvant chemotherapy start), with only a slightly higher frequency in the >50% Nectin 4 group (25.0% compared to each 16.7% for the 0% and ≤50% Nectin 4 groups).

Univariate Cox regression analyses (Table 3) and a Kaplan-Meier estimate indicate a negative impact of >50% Nectin 4 abundance on overall survival (HR 3.03, CI_95_ 1.37–6.68; *p* = 0.006; Figure 4D). Even after correction for age, FIGO stage, grade, and residual tumor mass in a multiple Cox regression analysis (Table 3), high Nectin 4 abundance showed an independent negative impact on overall survival (HR 3.62, CI_95_ 1.52–8.63; *p* = 0.004), illustrated as survival curves of the Cox regression model in Figure 4E. Adding CA-125 and chemotherapy mode to the multiple Cox regression model increased the HR for OS of high Nectin 4 expression to 4.64 (CI_95_ 1.52–14.13) (Appendix A). CA-125 showed no impact on OS and chemotherapy mode only univariately (HR for neodajuvant chemotherapy 2.46, CI_95_ 1.01–6.04, *p* = 0.049). Similar analyses for progression free survival (Appendix A) revealed a significant univariate impact (HR 2.60, CI_95_ 1.33–5.08; *p* = 0.005) which did not hold in the multiple Cox regression model (HR 1.92, CI_95_ 0.94–3.95; *p* = 0.074).

### 2.4. Network of Associations of the Nectin 4 Driven Expression Clusters with Omics Data and Biological Interpretation

Aim of this section was to examine the biological impact of the candidate hub-gene on tumor tissues, the microenvironment (i.e., peritoneal), and systemically, mainly on the immune reactions. Therefore Nectin 4 protein expression was correlated to omics and medium dimensional data derived from tumor tissues, the ascites, and the serum of the patients by various statistical approaches, like pathway, network, and gene-set analyses. Finally, the complex associations between these data and results–correlated to the Nectin 4 expression–were presented in a richly annotated network representation.

To examine the impact of high Nectin 4 expression on tumor cells, the local intraperitoneal microenvironment and systemic characteristics of patients, the trichotomized Nectin 4 score (0%, ≤50%, and >50%, respectively) was correlated to (i) gene expression data of isolated solid tumor cells (ovarian and peritoneal tumor masses) and free floating ascitic tumor cells (single and spheroids) [6,12], to (ii) immune cell compositions in tumors, ascites, and serum of patients (FACS and immunofluorescence staining data) [7], to (iii) cyto- and chemokines in ascites and serum [7], and to (iv) untargeted proteomics data from cell free ascites. In Table 4 these data and results are summarized.

Ad (i) Differential gene expression analysis revealed 908 significantly deregulated genes (FDR<5%; thereof 643 up-regulated in Nectin 4 high expressing tumors). Using 2003 deregulated genes (FDR < 10%) for a pathway analysis applying enrichment and topological information with 199 KEGG pathways revealed eight pathways significantly deregulated (FDR < 5%; all inhibited; Appendix A). The major histocompatibility complexes I (MHC I) and II (MHC II) were down-regulated in many of these inhibited pathways (Appendix A), therefore we assessed the expression of the single HLA genes in detail. In Appendix A all expressed MHC I and II HLA genes are shown, correlated to the Nectin 4 score. Five of seven MHC I and six of 13 MHC II associated genes were down-regulated (FDR < 20%) in Nectin 4 high expressing tumors. Whereas only two of 15 non-coding (pseudo- or antisense-) HLA genes were down-regulated (Appendix A).

Using a gene-set enrichment analysis with the 653 TCGA HGSOC expression data derived-clusters (*cf*. above and Figure 2) revealed 20 significantly deregulated clusters (FDR < 5%; thereof 12 up-regulated in Nectin 4 high expressing tumors; Appendix A). A further gene-set enrichment analysis with the gene-sets v6.2 from the Broad Institute [27] revealed 1828 of total 17,810 gene-sets significantly deregulated (FDR < 5%; 133 up-regulated in Nectin 4 high expressing tumors). The gene-sets “Chr22q12” and “Chr22q11” were five orders of magnitude more significant compared to all other gene-sets, indicating a chromosomal aberration at this region associated with Nectin 4 expression. To further examine this finding the log_2_ fold changes associated to the Nectin 4 score of all genes mapped to chromosome 22 were plotted ordered relative to their positions. And indeed, significantly reduced log_2_ fold changes can be seen between 22q11.21 and 22q12.3 chromosomal bands (Appendix A). Another evidence for copy loss of a chromosomal region correlated with the Nectin 4 score arose from the analysis of cluster *c1_496* (*cf*. Table 5 and Appendix A). Twenty out of 23 genes from this cluster map to chromosomal band 6q21 and two further genes to 6q16 and 6q22. An indeed, 6q is the most affected chromosomal loss region in ovarian cancer [4], indicating LOH as driving event during the co-association network generation from expression data and sub-cluster identification of *c1_496*. The expression of the cluster *c1_496* is negatively correlated to the Nectin 4 score indicating a positive correlation of 6q21 LOH frequency with Nectin 4 expression.

Furthermore, significantly deregulated genes were mapped onto four networks, (i) the TCGA expression derived co-association network (“TCGAnet”, a HGSOC specific co-expression network, *cf*. above), (ii) the STRING v10.0 network (“STRING”, a direct and functional (literature) driven protein-protein interaction network, [28], (iii) an experimentally verified protein-protein interaction network (“PPI”; CCSB Human Interactome database, HI-III, preliminary release 2.5, downloaded from http://interactome.dfci.harvard.edu), and (iv) a network of all connected KEGG pathways (KEGG super-pathway, “KEGG”, [29]), and the maximum scoring sub-networks with at least 30 nodes (20 for the PPI network) determined. In Appendix A these four sub-networks and in Appendix A eleven of the smaller (<30 nodes) significantly deregulated TCGA clusters are shown and in Table 5 some hub genes and the putative functions of these high-scoring sub-networks and clusters are summarized. Interestingly, seven out of the 110 genes present in the four high-scoring sub-networks were from the 263 genes mapping to bands q11 and q12 on chromosome 22 (Appendix A), which is a highly significant enrichment (*p* < 0.0001).

To correlate these sub-networks and clusters (represented by the first principal component (PC1) of the expressions of the network genes) with other omics and medium dimensional data from tumor tissues, the microenvironment (i.e., ascites), and systemically (i.e., blood) derived data (*cf*. Table 4), significant correlations between PC1s of the high-scoring sub-networks and clusters with all analytes were determined (FDR < 10%) and are shown as comprehensive network in Figure 5. The validity of this approach is exemplified by the revealed correlations of the PC1 of the *c1_358* TCGA cluster with—inter alia—eleven Kallikrein genes (including KLK6) with the soluble KLK6 abundance in ascites (bottom middle Figure 5) or of the PC1 of the PPI network (including BCAM) with the soluble BCAM abundance in ascites. Using this approach, sub-networks and clusters with similar functions were arranged close together even if the overlap of the sub-networks and clusters was rather small (Appendix A), like clusters *c1_747* and *c1_782* (Appendix A) or cluster *c1_315* and the TCGA- and KEGG-sub-networks (Appendix A, respectively).

Among the immune cell populations, mainly CD14 positive cells (e.g., monocytes) and several different natural killer (NK) cell populations (especially NK1 and NK2 [7,30]) in the tumor tissue, blood, and ascites were underrepresented in Nectin 4 positive tumors and were significantly correlated to many sub-networks and clusters (*cf.*
Figure 5).

### 2.5. NECTIN4 Dependent Expression Differences in HGSOC Cell Lines

To estimate functional expression differences between each three NECTIN4-high (Caov-3, OVCAR-3, and OVKATE) and NECTIN4-low (OV-90, ES-2, and TYK-nu) expressing HGSOC cell lines, publicly available RNA-sequencing data were used for gene expression analyses [25]. Of about 18k expressed genes, 53 genes were higher and four lower expressed (FDR < 5%) in NECTIN4-high cell lines (or 718 higher and 194 lower, respectively, with a 20% FDR cutoff) (*cf*. Table 4, Appendix A “sign_RNAseq_CLs” and Appendix A). Again, genes from the NECTIN4-cluster (*c1_143*) introduced in Figure 2F and Figure 4B were significantly enriched in high ranks, indicating a significant up-regulation of the complete cluster (Figure 6B). L1CAM was the most up-regulated single gene, known to be a negative prognostic factor in ovarian cancer [31,32] (Figure 6A), therefore fitting to the worse outcome of patients with high Nectin 4 expressing tumors in our study.

A Gene Ontology (GO) enrichment analysis of the 20% FDR up-regulated genes revealed “homophilic cell adhesion via plasma membrane adhesion molecules” as the most relevant *biologic process* GO-term and “negative regulation of MAPK cascade” for the down-regulated genes (Figure 6C). A SPIA pathway analysis revealed four (FDR < 5%; seven with FDR < 20%) activated pathways (Figure 6D and Appendix A “sign_KEGG_CLs”): “Basal cell carcinoma”, “Hippo signaling pathway”, “Breast cancer”, “Melanogenesis”, (“Arrhythmogenic right ventricular cardiomyopathy (ARVC)”, “Gastric cancer”, and “Cushing syndrome”) (Appendix A). In the ARVC pathway (Figure 6E) highly activated desmosome gene expressions but an inhibited connexin (gap junctions) gene expression and a shift from α-integrins (ITGAs) to β-integrins (ITGBs) were revealed.

## 3. Discussion

In search of new targets for which putative therapeutics have already been developed, a comprehensive analysis including various validation steps revealed new evidence for Nectin 4 as potential target for selected ovarian cancer patients. A clinically functional antibody against Nectin 4 has already been developed (enfortumab) and its relevance for epithelial cancers hypothesized [33], especially as antibody-drug combination (monomethyl auristatin E, MMAE, a small molecule microtubule-disrupting agent), known as enfortumab vedotin or ASG-22ME. This antibody-drug combination is already under investigation in a phase I clinical trial (clinicaltrials.gov identifier ASG-22CE-13-2) and first results were presented at the European Society for Medical Oncology (ESMO) 2016 congress. So far, in ovarian cancer only little is known about Nectin 4 expression and its impact on outcome. Derycke et al. showed that Nectin 4 is overexpressed in 51.5% of high grade serous ovarian cancer tissues but found no impact on survival in a heterogeneous population comprising different histologies (serous, mucinous, endometrioid, and clear cell), grades, and FIGO-stages in Kaplan-Meier estimates [34]. Aberrant expression of Nectin 4 was shown for several cancer entities, including ovarian cancer with comparable expression frequencies in primary serous tumors, i.e., 47%, not discriminating between high and low grade, and higher frequencies in metastases, i.e., 79%, in serous ovarian cancer metastases (of unknown origin, peritoneal local or “real” distant) [33]. In a study evaluating NECTIN4 expression by qPCR and ELISA in 39 ovarian cancer patients, 21 subjects with benign ovarian pathologies and 25 healthy controls, Nabih et al. [35] demonstrated increased NECTIN4 mRNA expression in 97.4% of the ovarian cancer samples. In human ovarian cancer cells the significance of Nectin 4 in cell-cell adhesion, proliferation, and migration has been demonstrated by over-expression and silencing experiments [36]. Artificial higher NECTIN4 expression compared to biological or silenced NECTIN4 expression in cell line models was associated with a more epithelial phenotype [36], fitting to our results that cell lines with biological higher NECTIN4 expression are from the miliary type [12], shown to exhibit high epithelial characteristics [6].

Further, growth of orthotopically implanted breast cancer cells in nude mice can be inhibited by blocking PVRL4 (NECTIN4)-driven cell-to-cell attachment with monoclonal antibodies against PVRL4 [37].

Further information about expression or impact on outcome in (high grade serous) ovarian cancer is not available. In tumor specimens of 5673 triple negative breast cancer patients Nectin 4 has been identified as cell surface biomarker [38]. Nectin 4 was overexpressed in triple negative breast cancer and basal breast cancer patients. Further, these findings were strongly correlated to NECTIN4 mRNA expression. In multivariate analysis high NECTIN4 mRNA expression has been identified as independent poor prognosis factor for metastasis-free survival. To test Nectin 4 as therapeutic target an monoclonal anti-Nectin 4 antibody drug (monomethyl auristatin-E; MMAE) conjugate was investigated in vitro and in vivo. In both, localized and metastatic triple negative and Nectin 4-positive breast cancer this therapy led to rapid and long-lasting regression in mice xenograft models. In hepatocellular carcinoma (HCC) Nectin 4 was over-expressed in 68% HCC tissues and positive Nectin 4 expression was significantly correlated with tumor size, status of metastasis, vascular invasion and tumor-node-metastasis stage. Nectin 4 expression was also associated with worse recurrence-free and overall survival, univariately and multivariately [39]. In gastric cancer higher Nectin 4 expression was found in cancerous compared to normal gastric tissues and was associated to unfavorable outcome [40].

In the data presented above, NECTIN4 was identified as hub-gene in a highly significant co-association cluster between samples from patients with miliary and non-miliary peritoneal tumor spread behavior. Subsequent protein expression analyses revealed positive Nectin 4 expression in about 53% of late stage HGSOC tissues but impact on survival was only seen in those 13% of cases with >50% of positive tumor cells (Table 3 and Figure 4). A comprehensive network based multi-omics integration revealed several interesting biological and immunological characteristics of high Nectin 4 expressing tumors (Figure 5): (i) expression of HLA genes–mainly from the MHC I complex–were negatively correlated to Nectin 4 expression, indicating an immunologic silencing strategy of these tumors, similar as we revealed in PD-L1 negative HGSOC tumors [41]; (ii) genes from chromosomal region 22q11-12 were highly over-represented and down-regulated in Nectin 4 high tumors (Appendix A), and this even seems to drive Nectin 4 associated high-scoring sub-networks (gene co-associations (TCGA clusters), functional (STRING) and experimentally verified PPI, and KEGG-super-pathway), i.e., seven out of 110 genes in these pathways were from this region, which is significantly over-represented (Appendix A; *p* < 0001). It is known that loss of chromosomal region 22q with 79% is the second most common chromosomal loss in HGSOC (after 16q with 80% [4], which seems also correlated to Nectin 4 expression, indicated by cluster *c1_496*, which is comprised of only genes mapping to 6q16-22 and expression of this gene-cluster was significantly negatively correlated to Nectin 4 expression); (iii) a cluster with 11 Kallikrein genes (*c1_358*; Appendix A) was positively correlated to Nectin 4 expression and also soluble Kallikrein 6 in cell free ascites (untargeted proteomics). Kallikreins are a subgroup of serine proteases capable of cleaving peptide bonds in proteins, and are associated with an increased metastatic behavior and unfavorable survival in HGSOC [42] and seem to be attractive targets for ovarian cancer treatment [43]; (iv) higher soluble protein abundance in ascites and gene expression of basal cell adhesion molecule (BCAM) was also identified, a receptor for the extracellular matrix protein laminin α5 (LAMA5), known to be expressed in ovarian cancer tissues [44]. The function of BCAM in cancer is largely unknown, only in KRAS-mutated colorectal cancer it was shown, that the axis BCAM-LAMA5 mediates the recognition between tumor cells and the endothelium in the metastatic spread [45], and (v) several immune cell population changes related to the clusters, systematically in blood and locally in ascites and the tumor, i.e., reduced CD14 positive cell concentrations (e.g., monocytes) in ascites (negatively correlated to clusters *c1_180* and *c1_363*), reduced concentrations of several natural killer cell populations in blood and the tumor [7] (negatively correlated to clusters c1_747, c1_782, c1_376, c1_335, and the high-scoring sub-networks PPI and STRING), and the cluster *c1_143* (with Nectin 4 as hub-gene) correlated with several immune modulating cytokines, e.g., in serum negatively with IL6, the soluble interleukin-6 receptor (sIL6Ra), MIF, and CCL20 and in ascites negatively with, CCL25, and prolactin. The immune-suppressive IL10 was positively correlated with the Nectin 4 cluster (*c1_143*). For further correlations see Figure 5.

Interestingly, ATG5 as member of the negatively to the Nectin 4 expression correlated *c1_496* cluster of 6q21 genes is a known key regulator of autophagosome formation. Next to autophagy activation, ATG5 is also involved in cell death progression [46]. Autophagy is abnormally activated during cancer cell metastasis and plays an important role in the maintenance of cancer cell viability [47]. Recently, in a mouse model ATG5 deficiency has been shown to be involved in tumorigenesis [48]. Downregulation of ATG5 transcription further leads to repression of macroautophagy activity, which promotes breast cancer cell metastasis [49]. In colorectal cancer tissues, ATG5 expression was also downregulated, which led to the assumption that ATG5 may function as a tumor suppressor [50].

Given the putative function of Nectin 4—Ca^2+^-independent cellular adhesion—the anti-Nectin 4 antibody might have additional effects in ovarian cancer besides being “the Trojan horse for an antineoplastic substance”, i.e., potential direct effect through inhibition of the attachment of tumor cells on the peritoneal wall, which might inhibit local peritoneal tumor spread directly. First underlying data can be derived from cell line models [36] or in vivo mouse models (breast cancer [37]).

Further functional analyses uncovering the direct impact of Nectin 4 and the other–mostly surface–proteins from the identified cluster, probably involved in cell-cell or cell-matrix attachment, are necessary to enlighten the functional impact in HGSOC, especially for peritoneal tumor spreading.

## 4. Material and Methods

### 4.1. Planar Filtered Gene Co-Association Network and Differential Cluster Expression Analyses

All analyses were conducted in R v3.5.0 [51] and CRAN or Bioconductor packages [52] (if not otherwise stated). Normalized read counts from RNA-sequencing data of 303 HGSOC tissues from The Cancer Genome Atlas (TCGA, https://cancergenome.nih.gov/) were downloaded via R-package TCGA2STAT v1.2 and filtered for genes with counts per million > 0.1 in halve of samples (20,501 genes). Counts were transformed and normalized with the function *voom* (“transforms count data to log_2_-counts per million (logCPM) [with offset 0.5], estimates the mean-variance relationship and uses this to compute appropriate observational-level weights. The data are then ready for linear modelling”) and the cyclic loess method (“Normalizes the columns of a matrix, cyclicly applying loess normalization to normalize each pair of columns to each other”. Cyclic loess is a non-linear but less aggressive normalization method compared to quantile normalization) implemented in the Bioconductor R-package limma v3.30.7 [53]. The gene co-association network was built with Tool for Inferring Network of Genes (TINGe v1.062) using mutual information criterion as association measure, a list of known transcription factors (*n* = 1672), and following parameters (converting final mutual information to correlation values): “−C c −w −v −p 1e−3 −e 0.2” [54]. The planar filtered (thus scale free small world) network was built from the TINGe network using the *calculate.PFN* function from R-package Multiscale Clustering of Geometrical Network (MEGENA v1.3.5-2) [55]. Multiscale clustering analysis (MCA) and multiscale hub analysis (MHA) was performed with *do.MEGENA* function from the same R-package, allowing clusters of sizes larger than ten and smaller than halve of nodes in the network (i.e., 10,250) [55]. The final list of clusters (*n* = 653) was used for gene-set analyses comparing miliary and non-miliary samples (as defined in [6]) from ascites (*n* = 22; A, single ascites tumor cells; S, spheroidal ascites tumor cell aggregates) and solid tumor tissues (*n* = 20; P, primary, i.e., ovarian, tumor mass; M, metastasis, i.e., peritoneal tumor mass) of 21 patients (twelve miliary and nine non-miliary) [6,12]. Gene-set analyses were performed with *mroast* [56] function for AS-samples and with *camera* [57] function for PM-samples (limma) using gene- and sample-weights from the *voomWithQualityWeights* function (limma) [58]. Raw *p*-values of both comparisons were combined by the Fisher’s method and corrected for multiple testing by the Benjamini-Hochberg (False Discovery Rate, FDR) method (R-package MetaDE v1.0.5). The same gene expression data and additional gene expression data of eight partly immortalized “normal” ovarian and tubal cell(-line)s were used for triplots (R-package ggtern v2.2.0) in Figure 1B, with calculated 272 miliary-versus-non-miliary gene signature (right edge) [6], 211 ovarian-versus-tubal origin gene signature (left edge) [12], and 212 epithelial-mesenchymal (EM) gene signature (bottom edge) [17] scores (always median first type up, i.e., miliary, ovarian, or epithelial, genes minus median second type up, i.e., non-miliary, tubal, or mesenchymal, genes). Values were scaled between zero and one before plotting with ggtern. Usually a false discovery rate (FDR) cutoff of 5% was chosen, but 10% (e.g., heatmap of differentially expressed genes, correlations of associations between different omics-results in Figure 4, or for some results in Table 4) or even 20% (e.g., pathway analyses for differences between Nectin 4 high and low cell lines) were also used in specific cases. The used FDR-cutoff is always indicated.

### 4.2. Cell Line Analyses

Ovarian cancer cell lines were analyzed as described in Auer et al. [12] and triplots plotted with Nectin 4 gene expression as third value (bottom edge), instead of the 212 epithelial-mesenchymal (EM) gene signature score. Cell lines were classified as of ovarian or tubal origin by the non-negative matrix factorization approach (NMF) using the 211 origin gene signature genes as described [12].

### 4.3. Overall Survival Meta-Analyses

Overall survival meta-analyses were performed with the curated ovarian cancer data sets from the Bioconductor R-package curatedOvarianData v1.12.0 [59]. In a first step, only samples of high grade (2/3) and late stage (FIGO stage III/IV) with serous histology were selected with further information for age and residual tumor mass after debulking surgery and used for univariate and multiple Cox-regression analyses (with age, FIGO-stage, and residual tumor mass as correcting factors) together with the dichotomized Nectin 4 cluster gene signature score (calculated as follows: median expression of all genes up in miliary, *n* = 31, minus median expression of all genes down in miliary, *n* = 4; dichotomized at the 25 percentile; Appendix A). Results are shown in forest plots and combined hazard ratios with 95% confidence intervals and combined *p*-values.

### 4.4. Immunohistochemistry of Nectin 4 and Impact on Overall Survival

Whole tissue sections from ovarian tumor tissue were employed for immunohistochemistry (IHC). Staining was performed as described previously [60] using a polyclonal antibody against Nectin 4 (dilution 1:500, cat no. Ab192033; Abcam, Cambridge, MA, USA) that had already been implemented successfully in tissue samples of solid tumors. Antigen heat retrieval was performed by microwaving the slides in EDTA (0.01 M, pH 8.0). The sections were incubated at 4 °C overnight with primary antibodies. Placental tissue sections were used as positive controls. For negative control ovarian and placenta sections were incubated in absence of the primary antibody. All quantitative assessments (H-scoring, 0, 1+, 2+, and 3+, and percentage positive cells) were done by visual scoring independently by two trained IHC analysts specialized in gynecology. The Cohen’s kappa coefficient (κ) was calculated for the inter-rater agreement according the H-score and discordant samples were looked at together to resolve the discrepancy. Percentage positive cells were averaged between both raters. Approval for this study was obtained by the ethical review board (Ethics Committee of the Medical University of Vienna: nos. 366/2003, 793/2011, and 1076/2018) and all patients signed an informed consent.

Table 2 shows the characteristics of the 90 patients included in the IHC and survival analyses (Table 3). Only patients with advanced stage, high grade serous EOC were included. Mean age at time of cytoreductive surgery was 60.0 years (SD 11.9 years). The median observation period for progression free survival was 56.8 months (IQR 27.3–62.9 months) and for overall survival 47.4 months (IQR 24.9–65.5 months), estimated from patients without events, respectively. Within the observation period, 40 (44.4%) patients died and 71 patients (78.9%) experienced tumor recurrence or progression.

Non-linear modelling of percent Nectin 4-positive cells with fractional polynomials Cox regression was performed with R-package mfp v1.5.2 [26] using *fp(percNECTIN4, df = 4)* together with age, FIGO stage, grade, and residual tumor. After dichotomization at >50% (*cf*. Figure 3A) progression free and overall survival was estimated by univariate and multiple Cox regression (*coxph()*-function from R). Kaplan-Meier estimates of univariate Cox regressions and survival curves of the multiple Cox regression model were plotted with R-function *plot(survfit(..))*, for the latter with all parameters averaged except the dichotomous Nectin 4 variable. *p*-values below 0.05 were considered as statistically significant.

### 4.5. Untargeted Proteomics of Cell Free Ascites

Cell free ascites samples were depleted using Pierce™ Top 12 Abundant Protein Depletion Spin Columns (Thermo Fisher Scientific, Waltham, MA, USA). Samples were depleted according to the protocol of the manufacturer. In-solution digestion of depleted samples was performed on 3 kDa MW cutoff filters (Nanosep with Omega membrane, Pall Austria Filter GmbH, Vienna, Austria). Total protein amount of 20 µg was used for digestion, which consists of protein reduction (dithiothreitol, Gerbu Biotechnik GmbH, Heidelberg, Germany), protein alkylation (2-iodoacetamide, Sigma-Aldrich, St. Louis, MO, USA) and protein digestion with a Trypsin/Lys-C Mix (MS grade; Promega Corporation, Madison, WI, USA) [61]. Untargeted proteomics analysis of digested peptide was conducted on a QExactive Orbitrap mass spectrometer (Thermo Fischer Scientific) coupled with a UltiMate 3000 RSLC nano System (Pre-column: Acclaim PepMap 100, C18 100 µm × 2 cm; Analytical column: Acclaim PepMap RSLC C18 75 µm × 50 cm; Dionex, Sunnyvale, CA, USA) [61]. A 54 min gradient from 1% to 40% solvent B was applied for chromatographic peptide separation. Solvent composition: A—7.9% water, 2% acetonitrile, 0.1% formic acid and B—97.9% acetonitrile, 2% water and 0.1% formic acid. The resolution on Orbitrap mass spectrometer was set to 70,000 for full MS scan and the MS2 scan was acquired at a resolution of 17,500, both at m/z 200. The MaxQuant 1.5.2.8 [62] software packet was used for protein identification and MS1 based label-free protein quantification (LFQ). The same criteria as described recently were applied for protein identification and MS1 based protein qualification [61]. Briefly, search criteria included a maximum of two missed cleavages and a maximal mass deviation of 5 ppm for peptide ions and of 20 ppm for fragment ions. Further, a minimum of two peptide identifications per protein (including one unique) was requested and an FDR of less than 0.01 was applied at both peptide and protein level.

### 4.6. Correlations to Omics and Other Data and Network Construction for Biological Interpretation

Differentially expressed genes associated with the Nectin 4 protein score (0, negative; 1, ≤50%; and 2, >50%) were assessed from raw RNA-sequencing reads [6,12] after cyclic loess normalization with function *voomWithQualityWeights* from R-package limma. Differentially abundance analyses of omics and other data (*cf*. Table 4) were performed also with limma after logarithmization (base 2) of the corresponding (normalized) read-outs (zeros were replaced by 0.1 before logarithmization). Differentially regulated KEGG-pathways were determined by a combined gene over-representation and perturbation analysis using Signaling Pathway Impact Analysis (SPIA), implemented in the Bioconductor R-package SPIA v2.32.0 [63], with 10,000 permutations and the normal inversion “norminv” *p*-value combination method. Correction for multiple testing was done by the False Discovery Rate (FDR) method (Benjamini-Hochberg). Significant pathways were illustrated with the Bioconductor R-package pathview v1.20.0 [64]. Over-representation analyses for clusters was performed with Quantitative Set Analysis for Gene Expression (QuSAGE), implemented in the R-package qusage v2.14.0 [65]. Gene-set analyses was performed with the Molecular Signatures Database (MSigDB) v6.2 [27] and the function camera [57] from R-package limma (independent variable: the trichotomized Nectin 4 score). Finally, the R-package dnet v1.1.4 [66] was used to find high-scoring sub-networks using the *p*-values from the differentially gene expression analysis above, either using the STRING v10 database of known (including literature-based) and predicted protein-protein interactions (“STRING”; [67]) or the more selective–only experimentally verified physical protein-protein interactions–CCSB Human Interactome–database (“PPI”; HI-III, preliminary release 2.5, downloaded from http://interactome.dfci.harvard.edu), a network of all connected KEGG pathways (KEGG super-pathway, “KEGG”, [29]), and the complete scale-free small-world co-association gene expression network (“TCGA”; *cf*. Figure 2). *p*-value cutoff for the heuristically finding of the high-scoring sub-networks were optimized to get at least 30 (for the PPI network, 20) nodes.

For the correlations shown in Figure 4, the four high-scoring subnetworks (STRING, PPI, KEGG, and TCGA) and all significant with Nectin 4 associated TCGA-clusters with less than 50 nodes (*n* = 12) were used. All genes with available gene expression data from these clusters were summarized over all samples using the first principal component (PC1) of a principal component analysis (PCA) using the *prcomp(center = TRUE, scale = TRUE)* function from R. In Figure 4 (right border) a boxplot of all 16 percentage of explained variations (PEVs) of all PC1s are shown. Using these PC1s as independent variables, all omics and other data sets were screened for significant correlations (using limma) and all significant correlations (FDR < 10%) were collected (including their orientation, either positively or negatively correlated). These correlations were than plotted in a network as colored edges (blue, negatively; red, positively correlated) between all PC1s of the four high-scoring sub-networks and twelve significant clusters and all analytes (immune cell populations, chemo/cytokines, and cell-free proteins). In Table 4 these data and results are summarized. In addition, the correlations between the 16 high-scoring sub-networks and clusters were assesses by graphical Gaussian modelling (GGM) using R-package GGMselect v0.1-12.1) [68]. The principle behind GGM is to use partial correlations as a measure of independence of any two PC1s which allows for distinguishing direct from indirect correlations. Raw RNA-sequencing data are available at ArrayExpress with accession no. E-MTAB-7674.

## 5. Conclusions

This study identified Nectin 4 as correlated to extensive, miliary, peritoneal tumor spread in HGSOC patients and as significantly associated with survival. A clinically functional antibody (antibody-drug combination) against Nectin 4 has already been developed, its relevance for epithelial cancers hypothesized and first encouraging antitumor activities presented in metastatic urothelial cancers. Our study provides comprehensive biological data highlighting that this targeted therapy might be an interesting approach in selected patients with high grade serous ovarian cancer.

## Figures and Tables

**Figure 1 cancers-11-00698-f001:**
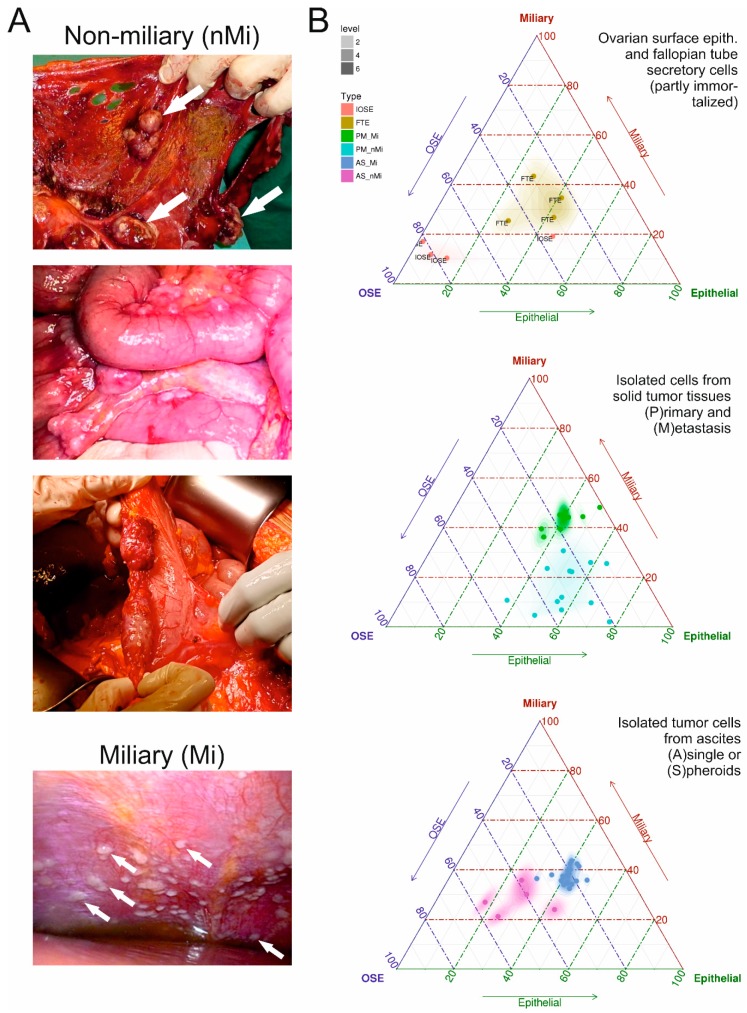
(**A**) Exemplarily presentations of miliary (left) and non-miliary (right) peritoneal tumor spread in high grade serous ovarian cancer. (**B**) Ovarian surface (OSE, pink) and fallopian secretory (FTE, ochre) epithelial cells (left triplot), isolated tumor cells from solid tumors (middle triplot), and from ascites (right triplot) characterized according three gene signatures: a 272 miliary-versus-non-miliary gene signature (right edge) [6], a 211 ovarian-versus-tubal origin gene signature (left edge) [12], and a 212 epithelial-mesenchymal (EM) gene signature (bottom edge) [17] and colored according tumor spread characteristic (green and blue, miliary; cyan and violet, non-miliary). For details see Material and Methods.

**Figure 2 cancers-11-00698-f002:**
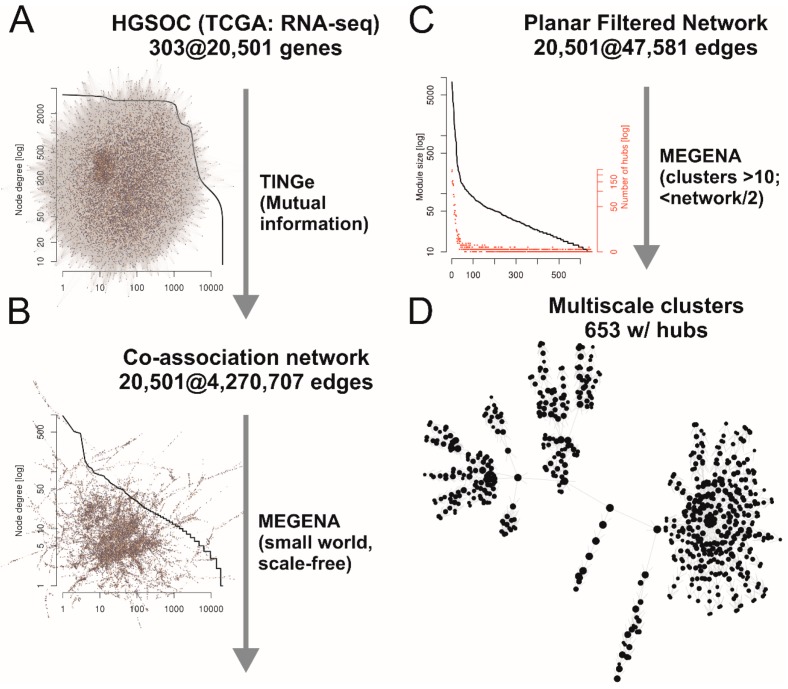
Outline of experimental procedures and results. (**A**) Gene co-association network of TCGA RNA-sequencing data of 303 HGSOC samples built with Tool for Inferring Networks of Genes (TINGe). The plot shows the node-degrees for all nodes (sorted according node-degree) in a log_10_-log_10_-scale. (**B**) Planar filtered–small world scale free–network (MEGENA, R-package). The plots shows the same as in (**A**), proving the scale-freeness (i.e., power law distribution). (**C**) Sub-networks (multiscale clusters) with corresponding hub-genes (MEGENA). The plot shows numbers of genes (black) and corresponding numbers of hub genes (red) in the clusters in a semi-log_10_ scale (y-axes). (**D**) Hierarchical network of 653 clusters, comprised of 10 to 6869 genes, used as gene-sets for gene-set analyses between samples from patients with miliary and non-miliary tumor spread. (**E**, below) RNA-sequencing samples from isolated cells of solid tumor tissues or from ascites from patients with intrasurgically defined tumor spread characteristic (miliary versus non-miliary). (**F**) Cluster *c1_143*, comprised of 43 genes and Nectin 4 as single hub-gene, highly significantly up-regulated in miliary tumor cells. Boxplots represents Nectin 4 expressions in corresponding tumor cells (solid tissues, blue box or ascites, yellow box) of non-miliary (green) or miliary (red) spread. Node colors of the cluster represent up (red) or down (green) regulation in miliary (left, tumor cells from ascites and right, isolated tumor cells from solid tumor tissues). (**G**) Nectin 4 cluster predictor, calculated from the genes always up in miliary (median) minus genes always down in miliary (median) and dichotomized at the 25 percentile (*cf*. main text). (**H**) Forest plots and combined hazard ratios and *p*-values of six independent patient cohorts for the dichotomized Nectin 4 cluster predictor in univariate Cox-regressions (left) or multiple Cox-regressions, corrected for age, FIGO stage, and residual tumor mass (right). Samples were pre-selected for high grade (2/3), late FIGO stage (III/IV) and serous histology.

**Figure 3 cancers-11-00698-f003:**
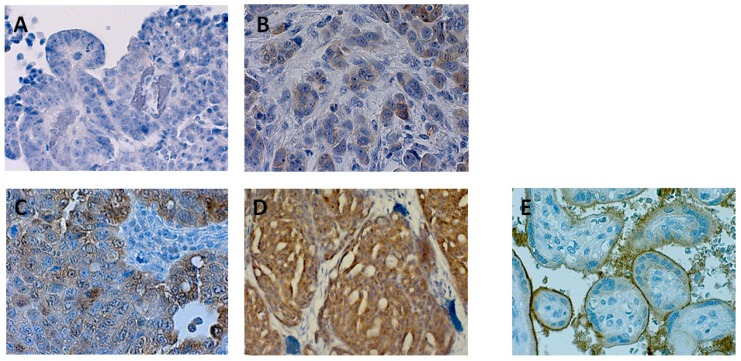
Representative pictures of immunohistochemical stainings of Nectin 4. (**A**) negative (nMi), (**B**) 20% + (n.d.), (**C**) 40% ++ (Mi), (**D**) 90% +++ (Mi), (**E**) positive control (placenta) (400× magnification). (nMi, non-miliary; Mi, miliary; n.d., not determined.).

**Figure 4 cancers-11-00698-f004:**
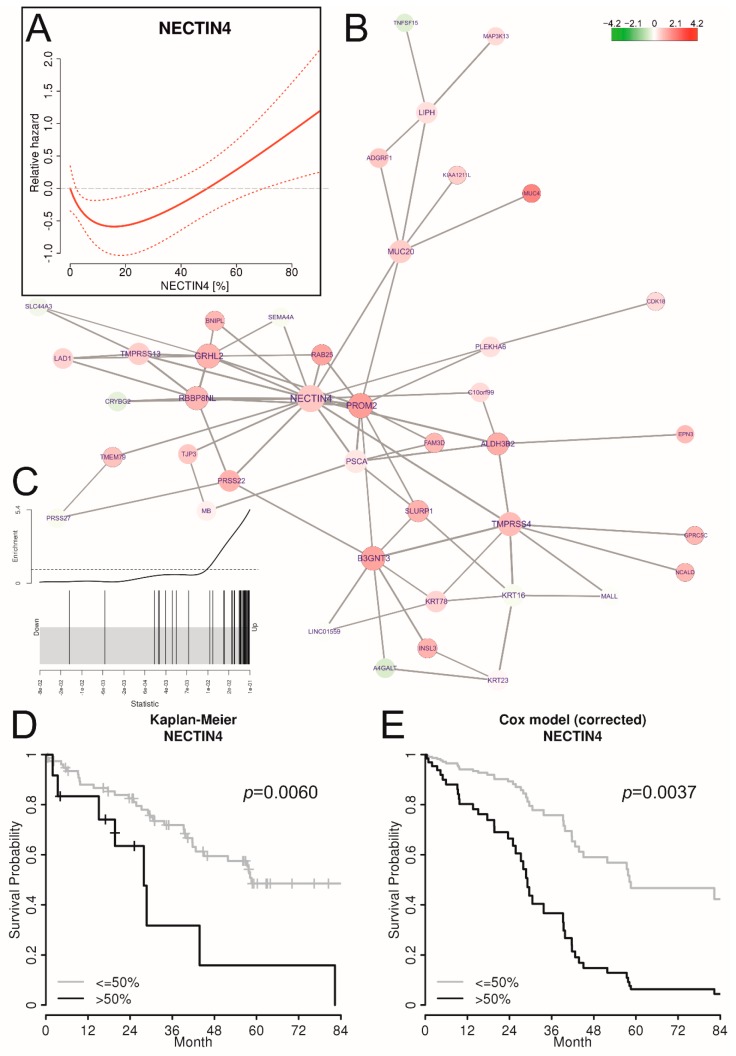
Validation of the impact of Nectin 4 protein expression on overall survival and correlation to Nectin 4 cluster (*c1_143*) expression. (**A**) Non-linear correlation of percentage Nectin 4 positive cells (*x*-axis) with relative hazard for death (*y*-axis) corrected for age, FIGO stage, and residual tumor mass, as estimated by fractional polynomials Cox regression. (**B**) Correlations of gene expressions of the *c1_143* cluster with NECTIN4 as hub-gene with Nectin 4 protein abundances (Nectin 4 score) in corresponding tumor tissues (colors represent log_2_ fold-changes, FCs). (**C**) Barcode enrichment plot showing the ranked statistics of log_2_ FCs from subfigure B. (**D**) Kaplan-Meier estimate for overall survival dichotomizing Nectin 4 percentages following the cutoff determined in subfigure A (>50% versus ≤50%). (**E**) Survival curves of the multiple COX regression model (*cf*. Table 3), dichotomized as in subfigure D. As these survival curves represent a multiple Cox model, no censored patients are indicated.

**Figure 5 cancers-11-00698-f005:**
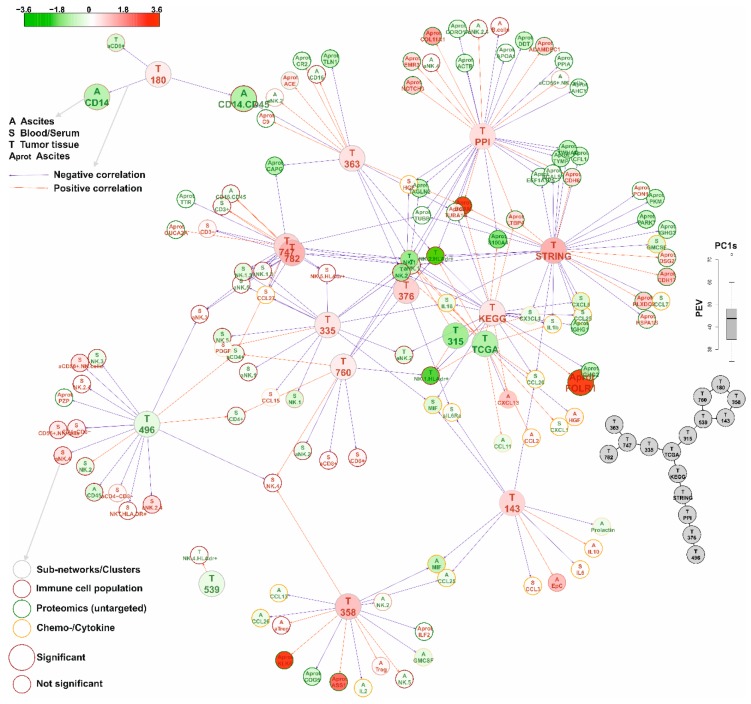
A zoomable version of this network is provided as pdf in the Supplement as Appendix A. Network of significant Nectin 4 correlated gene-expression co-association clusters (*cf*. Table 5) correlated to other omics (proteomics), FACS (immune cell populations) and Luminex (cyto/chemokines) data (*cf*. Table 4). The gene-expression co-association clusters are represented by their first principal components (of all gene expression values), PC1s. The distribution of the percentages of explained variations (PEVs) of these PC1s is shown at the right border. Below a network of the correlations of the PC1s of the gene-expression co-association clusters is shown as estimated by graphical Gaussian modelling (GGM). The main network shows all significant (FDR < 10%) correlations of the PC1s of the clusters with all other data. Edge colors represent the direction of the correlation (red, positive; blue, negative correlation), the color of the nodes indicates the log_2_ fold-changes (FCs) of the analytes/cell-populations with the percentages of Nectin 4 positive cells (Nectin 4 score), and the size of the nodes indicates if these FCs are significant (large, yes; small, no). The type and source of the analytes/cell-populations is coded in the node-label and node border color, respectively. Analytes and cell-populations are given as node-labels (for complete lists of analyzed factors *cf*. Table 4 and Appendix A). A comprehensive legend is shown on the left border of the figure.

**Figure 6 cancers-11-00698-f006:**
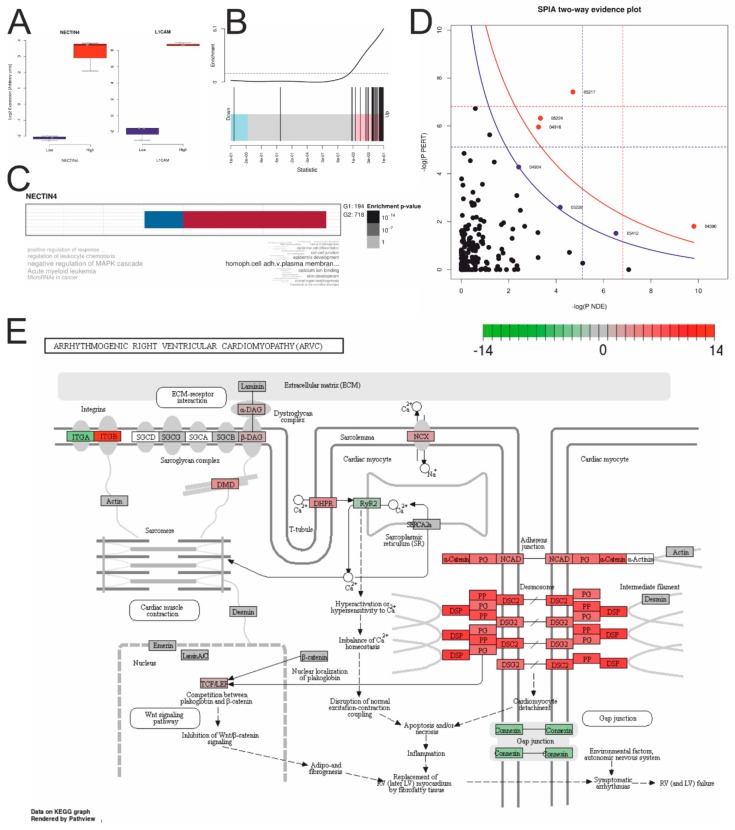
Gene expression differences between each three NECTIN4 high (Caov-3, OVCAR-3, and OVKATE) and three low (OV-90, ES-2, and TYK-nu) expressing HGSOC cell lines (*cf*. Table 4). (**A**) Boxplots of NECTIN4 and L1CAM expression. (**B**) Barcode enrichment plot showing the ranked statistics of log_2_ FCs according the *c1_143* cluster (*cf*. Figure 4B). (**C**) Gene Ontology enrichment analysis with down- (G1) and up-regulated (G2) genes. (**D**) SPIA evidence plot of significantly deregulated KEGG pathways (blue line: cutoff FDR < 20%; for details *cf*. Appendix A). (**E**) KEGG pathway “Arrhythmogenic right ventricular cardiomyopathy” significantly activated in NECTIN4 high expressing cell lines. Node colors represent the log_2_ fold changes (Appendix A “sign_KEGG_CLs”; for the other activated KEGG pathways *cf*. Appendix A).

**Table 1 cancers-11-00698-t001:** Table of sub-clusters differentially expressed in tumor cells of patients with miliary peritoneal tumor spread compared to patients with non-miliary peritoneal tumor spread (AS, ascites tumor cells; PM, solid tumor tissues). In bold the Nectin 4 cluster.

Cluster	N Genes	Direction AS	P (AS)	Direction PM	P (PM)
c1_160	37	Up	3.32 × 10^−^^1^	Up	3.92 × 10^−^^14^
c1_32	165	Up	1.00 × 10^−^^5^	Up	3.62 × 10^−^^9^
**c1_143**	**38**	**Up**	**1.00 × 10^−^^5^**	**Up**	**7.98 × 10^−^^9^**
c1_400	22	Up	1.00 × 10^−^^5^	Up	2.26 × 10^−^^8^
c1_297	86	Down	4.00 × 10^−^^2^	Down	9.31 × 10^−^^12^
c1_42	115	Up	6.51 × 10^−^^3^	Up	9.22 × 10^−^^11^
c1_54	72	Down	9.47 × 10^−^^1^	Down	7.50 × 10^−^^13^
c1_247	16	Up	1.00 × 10^−^^5^	Up	1.06 × 10^−^^7^
c1_39	96	Up	4.49 × 10^−^^2^	Up	2.66 × 10^−^^11^
c1_75	69	Up	1.00 × 10^−^^5^	Up	2.28 × 10^−^^7^
c1_134	30	Up	1.26 × 10^−^^1^	Up	8.37 × 10^−^^11^

Static plots (with regulation information, red, up in miliary, green, down in miliary, each with gene names labelled if significant and darker colored and always name-labelled, if hub genes) and interactive network-representations with genes (nodes) linked to GeneCards (http://www.genecards.org/) of the first three Clusters are linked in Appendix A. Bold, selected cluster with Nectin 4 as hub-gene.

**Table 2 cancers-11-00698-t002:** Relationship between clinicopathological parameters and Nectin 4 expression in 90 high-grade late-stage serous ovarian cancer patients.

Clinicopathologic Characteristics	*n* = 90	Nectin 4 Abundance			*p* (Overall)	*p* (Trend)
		Negative (*n* = 42)	≤50% (*n* = 36)	>50% (*n* = 12)		
Tumor stage ^1^					0.257	0.156
FIGO III	69	34 (81.0%)	28 (77.8%)	7 (58.3%)		
FIGO IV	21	8 (19.0%)	8 (22.2%)	5 (41.7%)		
Histological grade					0.090	0.454
G2	18	8 (19.0%)	10 (27.8%)	0 (0.0%)		
G3	72	34 (81.0%)	26 (72.2%)	12 (100.0%)		
**Residual disease**					**0.027**	**0.020**
**R0**	**40**	**23 (54.8%)**	**16 (44.4%)**	**1 (8.3%)**		
**R1**	**48**	**18 (42.9%)**	**19 (52.8%)**	**11 (91.7%)**		
**Missing**	**2**	**1 (2.3%)**	**1 (2.8%)**	**0 (0.0%)**		
Age, diagnosis [y] ^2^	60.0 (11.9)	58.2 (12.5)	61.1 (11.3)	62.6 (11.5)	0.398	0.181
Progression free survival					0.120	**0.037**
Without progression	19	13 (31.0%)	5 (13.9%)	1 (8.3%)		
With progression	71	29 (69.0%)	31 (86.1%)	11 (91.7%)		
Status					0.235	0.315
Alive	50	24 (57.1%)	22 (61.1%)	4 (33.3%)		
Died	40	18 (42.9%)	14 (38.9%)	8 (66.7%)		

^1^ FIGO, International Federation of Gynecologists and Obstetricians; ^2^ Mean (standard deviation). Bold, statistically significant without correction for multiple testing.

**Table 3 cancers-11-00698-t003:** Univariate and multiple overall survival analyses in 90 patients with late stage (FIGO III/IV) high grade serous ovarian cancer.

Overall Survival						
Cox Regression Analyses	Univariate ^1^			Multiple ^2^		
	HR	CI_95_	*p*	HR	CI_95_	*p*
**Age (decades)**	**2.25**	**1.61–3.16**	**<0.001**	**2.37**	**1.68–3.33**	**<0.001**
FIGO stage (IV vs. III)	1.59	0.79–3.20	0.198	1.55	0.75–3.20	0.240
Histological grade (G3 vs. G2)	0.85	0.41–1.79	0.672	0.50	0.22–1.14	0.099
Residual tumor (R1 vs. R0)	1.71	0.91–3.24	0.096	1.61	0.84–3.09	0.155
**Nectin 4 (>50% vs.** **≤50%) ^3^**	**3.03**	**1.37–6.68**	**0.006**	**3.62**	**1.52–8.63**	**0.004**

^1^ Univariate Cox-regression; ^2^ Multiple Cox-regression analysis; HR, Hazard Ratio; CI_95_, 95% confidence interval; ^3^ The optimal cutoff was assessed by non-linear modeling of the Nectin 4 impact on OS by fractional polynomials Cox regression estimation correcting for known clinicopathologic factors age, FIGO stage, grade, and residual tumor mass after debulking surgery. In Figure 4A the corrected relative hazard against the percentage of Nectin 4 positive tumor cells is shown indicating a negative impact on OS in tumors with >50%, and therefore this cutoff, >50% versus ≤50%, was used for outcome analyses; Bold, statistically significant.

**Table 4 cancers-11-00698-t004:** Significant genes, proteins, and immune cell populations associated with the Nectin 4 score and significant correlations between clusters and these factors; and significant genes and pathways associated with NECTIN4 expression in HGSOC cell lines (n.a., not applicable).

Analytes Type	Tissue	N (Analytes)	FDR < 5% (<10%)	UP	Correlations FDR < 0%	Positive	Name of Table ^1^	Reference
RNA sequencing	Floating (ascites) and solid tumor cells	34,034 genes	908	643	n.a.		sign_RNAseq	[6,12]
--KEGG pathways		199	8 (9)	0 (0)	n.a.		sign_KEGG	-
--TCGA clusters		653	20 (58)	12 (42)	n.a.		sign_TCGA	-
--Gene-sets		17,810	1828	133	n.a.		sign_GS	-
Immune cells (FACS)	Ascites	43 cell populations	0	0	9	7	sign_A_FACS	[7]
	Blood	43 cell populations	0 (5)	(2)	44	10	sign_B_FACS	[7]
	Solid tumor tissue	43 cell populations	0	0	37	8	sign_T_FACS	[7]
Immune cells (IF)	Ascites	8 cell populations	0 (2)	0	8	4	sign_A_IF	[7]
Chemo/cytokines	Ascites	56	0	0	15	4	sign_A_Lum	[7]
	Serum	56	0	0	35	6	sign_S_Lum	[7]
Proteomics	Ascites	852	3 (4)	1 (2)	65	24	sign_A_Prot	-
RNA seq.	Cell lines	18,054 genes	57 (912) ^2^	53 (718) ^2^	n.a.		sign_RNAseq _CLs	[25]
KEGG pathways	Cell lines	199	4 (7) ^2^	4 (7) ^2^	n.a.		sign_KEGG _CLs	-

^1^Appendix A—worksheets. ^2^ FDR < 20%.

**Table 5 cancers-11-00698-t005:** List of Nectin 4 associated significantly deregulated HGSOC specific gene co-association clusters (derived from gene expression data of 303 HGSOC patients (TCGA), *cf*. Figure 2) with corresponding hub-genes and putative functions. The correlations with the Nectin 4 expression are indicated by UP/DOWN and colors red/green, respectively. These clusters were used for multi-omics correlation analyses shown in Figure 5 (*cf*. Table 4).

Network	Direction/ Hub Genes	Network Function/Gene Names	Description
c1_143	UP	Epithelial cell development; Adhesion	
	NECTIN4	Nectin cell adhesion molecule 4	involved in cell adhesion through trans-homophilic and -heterophilic interactions
	PROM2	Prominin 2	localizes to basal epithelial cells and may be involved in the organization of plasma membrane microdomains
	GRHL2	Grainyhead-like protein 2 homolog	primary neurulation and in epithelial development
	B3GNT3	UDP-GlcNAc:BetaGal Beta-1,3-N-Acetylglucosaminyltransferase 3	L-selectin ligand biosynthesis, lymphocyte homing and lymphocyte trafficking
c1_180	UP	Neuronal	
	PYCR2	Pyrroline-5-carboxylate reductase family	mutations identified as cause of a unique syndrome characterized by postnatal microcephaly, hypomyelination, and reduced cerebral white-matter volume
	MRPL55	Mammalian mitochondrial ribosomal proteins	protein synthesis within the mitochondrion
	SNAP47	Synaptosome Associated Protein 47	syntaxin binding and SNAP receptor activity
c1_363	UP	Apoptosis and energy metabolism (Mitochondrion)	
	TIMM50	translocase of inner mitochondrial membrane 50	maintaining membrane permeability barrier, knockdown of this gene results in the release of cytochrome c and apoptosis
	MRPS12	Mitochondrial Ribosomal Protein S12	key component of the ribosomal small subunit, controls the decoding fidelity and susceptibility to aminoglycoside antibiotics
	ECH1	Enoyl-CoA Hydratase 1	essential to metabolizing fatty acids to produce both acetyl CoA and energy
c1_747	UP	G protein signal-transducing/-1 frameshifting at translation	
	PNMA3	paraneoplastic Ma antigen	present in sera from patients suffering of paraneoplastic neurological disorders, promotes -1 frameshifting
	GPRIN2	G Protein Regulated Inducer of Neurite Outgrowth	interacted specifically with G-alpha-o and G-alpha-z bound to GTP-gamma-S and GDP-AlF4(−)
c1_782	UP	Epithelial proliferation; Neuronal?	
	PNMA3	paraneoplastic Ma antigen	present in sera from patients suffering of paraneoplastic neurological disorders, promotes-1 frameshifting
	KCNIP3	Potassium Voltage-Gated Channel Interacting Protein 3	Calsenilin, member of the family of voltage-gated potassium (Kv) channel-interacting proteins, belong to the neuronal calcium sensor family of proteins
	RSPO4	R-spondin 4	induced epithelial proliferation
c1_335	UP	Transcription regulation; Cell death	
	BEX3	Brain Expressed X-Linked 3	role in the pathogenesis of neurogenetic diseases
	TCEAL8	Transcription Elongation Factor A Like 8	modulate transcription in a promoter context-dependent manner.
c1_376	UP	Transcriptional regulation	
	ZNF574	Zinc finger protein 574	transcriptional regulation
	ZNF526	Zinc finger protein 526 (paralog to ZNF574)	highest expression in ovary
	GSK3A	Glycogen synthase kinase-3 alpha	type 2 diabetes, control of glucose homeostasis, Wnt signaling and regulation of transcription factors and microtubules, by phosphorylating and inactivating glycogen synthase
	BCAM	Basal cell adhesion molecule	may play a role in epithelial cell cancer; was shown to be overexpressed in ovarian carcinomas; BCAM-AKT2 fusion gene in 7% HGSOC cases
c1_760	UP	Transcriptional regulation	
	PHKG2	Phosphorylase Kinase Catalytic Subunit Gamma 2	mediates the neural and hormonal regulation of glycogenolysis by phosphorylating and thereby activating glycogen phosphorylase
	TBC1D10B	TBC1 Domain family, member 10B	Small G proteins of the RAB family
	7 ZNF genes	ZNF785, ZNF764, ZNF768, ZNF689, ZNF747, ZNF48, ZNF668	ZNF689: conferred anchorage-independent cell growth
c1_496	DOWN	Autophagy and splicing	20/23 genes from chromosomal band 6q21 and 2 further from 6q16 and 6q22
	CDC40	Cell Division Cycle 40	essential for the catalytic step II in pre-mRNA splicing process
	ATG5	Autophagy related 5	key regulator of autophagy mutations in the Atg5 gene have also been linked with prostate, gastrointestinal and colorectal cancers as ATG5 plays a role in both cell apoptosis and cell cycle arrest
c1_358	UP	11 KLK genes; Aberrant levels of kallikrein-related peptidases have been linked to cancer cell proliferation, invasion and metastasis	
	KLK10	Kallikrein-10	may play a role in suppression of tumorigenesis in breast and prostate cancers; shorter OS in HGSOC
	KLK6	Kallikrein-6	carcinogenesis, involvement in Alzheimer’s disease; shorter OS in HGSOC
c1_315	DOWN	Expressed on cells specialized for antigen presentation	
	CD1A-C; CD1E	Group 1 Thymocyte antigens	CD1 family of transmembrane glycoproteins, which are structurally related to the major histocompatibility complex (MHC) proteins; present lipid and glycolipid antigens to T cells
	FCER1A	Fc fragment of IgE, high affinity I, receptor for; alpha polypeptide	
KEGG	UP	Growth factors and cAMP -> MAPK signaling	
	EGFR	Epidermal Growth Factor Receptor	initiates several signal transduction cascades, i.e., the MAPK, Akt and JNK pathways, leading to DNA synthesis and cell proliferation.
	MAPK1	Mitogen-Activated Protein Kinase 1	intracellular signaling network that regulates many cellular machines, including the cell cycle machinery and autocrine/paracrine factor synthesizing machinery
	PRKACB	Protein Kinase CAMP-Activated Catalytic Subunit Beta	cAMP signalling towards the MAPK complex
	CREB5	CAMP Responsive Element Binding Protein 5	binds to CRE as a homodimer or a heterodimer with c-Jun or CRE-BP1, and functions as a CRE-dependent trans-activator
TCGAnet	DOWN	Immune cell infiltration?	
	CD53	Tetraspanin-25	expressed from several immune cells, B- and T-cells, monocytes, neutrophils, and NK cells
	HAVCR2	Hepatitis A virus cellular receptor 2	HAVCR2 is an immune checkpoint mediate the CD8+ T-cell exhaustion
	GLIPR1	Glioma pathogenesis-related protein 1	tumor suppressor
	PTPRC	CD45; receptor-type tyrosine-protein phosphatase C	pan immune cell marker
	GPR183	G protein-coupled receptor 183	expressed in B cells
STRING	UP	MAPK signaling and insuline pathway	
	MAPK1	Mitogen-Activated Protein Kinase 1	intracellular signaling network that regulates many cellular machines, including the cell cycle machinery and autocrine/paracrine factor synthesizing machinery
	CRKL	Crk-like protein	oncogene; participates in the Reelin signaling cascade downstream of DAB1
	AKT2	RAC-beta serine/threonine-protein kinase	oncogene; amplified and overexpressed in primary ovarian tumors
	CREB5	CAMP Responsive Element Binding Protein 5	binds to CRE as a homodimer or a heterodimer with c-Jun or CRE-BP1, and functions as a CRE-dependent trans-activator
PPI	UP	Transcription and splicing	
	TSGA10IP	Testis specific 10 interacting protein	
	MDFI	MyoD family inhibitor	transcription factor that negatively regulates other myogenic family proteins
	MEOX2	Homeobox protein MOX-2	transcription factor
	TFIP11	Tuftelin-interacting protein 11	associated with RNA and may play a role in splicing
	BCAM	basal cell adhesion molecule	overexpressed in ovarian carcinomas in vivo and upregulated following malignant transformation
	MAPK1	Mitogen-Activated Protein Kinase 1	intracellular signaling network that regulates many cellular machines, including the cell cycle machinery and autocrine/paracrine factor synthesizing machinery
	CREB5	CAMP Responsive Element Binding Protein 5	binds to CRE as a homodimer or a heterodimer with c-Jun or CRE-BP1, and functions as a CRE-dependent trans-activator

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
