# Peer review of "NECTIN4 (PVRL4) as Putative Therapeutic Target for a Specific Subtype of High Grade Serous Ovarian Cancer—An Integrative Multi-Omics Approach"

_cancers, 2019, doi:10.3390/cancers11050698_

Round 1
Reviewer 1 Report
This manuscript describes NECTIN4 as a potential target for therapeutics in the treatment of ovarian cancer. The approach to the identification of key gene sets in military and non-miliary ovarian cancers is thorough. The rationale for focus on NECTIN4 and the potential for more rapid clinical translation is sound. In addition to the highlighted findings the data present many other potential targets for investigation. I have no significant questions or criticism related to the experimental approach or data presentation.
Author Response
Nothing.
Reviewer 2 Report
In this manuscript, the authors use gene expression data to identify hub genes associated with the peritoneal spread patterns in patients with high grade serous ovarian cancer. The analysis identified Nectin 4. Using immunohistochemistry, they show that a high percentage of Nectin 4 is associated with increased residual disease and worse overall survival. Based on these results, the authors proposed that antibody-based therapies targeting Nectin 4 may be useful for the treatment of HGSOC. The authors also analyze gene and protein expression, and immune cell populations that correlate with Nectin 4 expression levels. However, as these parameters are not validated, this information is of limited significance.
Specific Comments:
The correlation between IHC staining for Nectin 4 and survival is one of the more important findings of this paper. It would be helpful to provide representative pictures of Nectin 4 high and Nectin 4 low samples, as well as controls.
The text on many figures is too small to be readable, particularly Fig. 1B, 2H, and 4.
Author Response
Specific Comments:
The correlation between IHC staining for Nectin 4 and survival is one of the more important findings of this paper. It would be helpful to provide representative pictures of Nectin 4 high and Nectin 4 low samples, as well as controls.
The Figure with representative pictures of Nectin 4 stainings is moved from the Supplement to the main manuscript (now Fig. 3).
The text on many figures is too small to be readable, particularly Fig. 1B, 2H, and 4.
Figures 1 and 2 were increased (Fig. 2 was split and one subgraph moved to the supplement). The font of Figure 4 (now Fig. 5) could not be increased, only changed to bold, due to graphical constrains and the resolution of the picture increased. Additionally this Figure is provided as zoomable pdf in the Supplement now (remarked also in the Figure legend).
Reviewer 3 Report
In their study, the authors used multi-omics approach to investigate the expression of various genes in miliary and non-miliary ovarian cancer and the correlation with Nectin 4 protein expression. They found that High Nectin 4 protein expression was associated with reduced expression of HLA genes, higher BCAM , higher Kallikrein gene and protein expressions; and v and reduced CD14 positive cells and reduction different natural killer cell population. They concluded that anti-Nectin 4 antibody with a linked be a candidate for a targeted therapy in patients with extensive peritoneal involvement.
This work is interesting but suffer some defects as shown in the following:
1-The last paragraph in introduction is too long, the aim of study should be simplified in a few clear sentences.
2-In table 5, there is down regulation of ATG5 in cancer cells. It will be better to add a few sentences in discussion indicating the relationship between autophagy and miliary ovarian cancer and Nectin 4 EXPRESSION.
3- Regarding the immunohistochemistry of Nectin 4 in supplementary figure one, which type of ovarian cancer expressed high levels of this protein?
4- Did the authors investigate the expression of Nectin 4 by Western blot in miliary and non-miliary ovarian cancer?
Author Response
1-The last paragraph in introduction is too long, the aim of study should be simplified in a few clear sentences.
The paragraph is simplified and shortened now.
2-In table 5, there is down regulation of ATG5 in cancer cells. It will be better to add a few sentences in discussion indicating the relationship between autophagy and miliary ovarian cancer and Nectin 4 EXPRESSION.
A thorough analysis of the sub-cluster c1_496 which includes ATG5, revealed that this cluster contains exclusively genes from chromosomal region 6q (mostly 6q21), which is the most frequent chromosomal loss region in ovarian cancer. We think therefore that the negative correlation of Nectin 4 expression with the expression of genes from this cluster is driven by this LOH and not by ATG5 directly. We added this information to the results section, to Table 5 and to the discussion. Nevertheless, a short paragraph of ATG5 function in metastasizing is also added to the discussion.
3- Regarding the immunohistochemistry of Nectin 4 in supplementary figure one, which type of ovarian cancer expressed high levels of this protein?
The Figure is now moved to the main manuscript (Fig. 3; due to a suggestion of another reviewer) and the type (miliary or non-miliary) is added as information to the legend. All are of course high grade serous.
4- Did the authors investigate the expression of Nectin 4 by Western blot in miliary and non-miliary ovarian cancer?
No, we only assessed the expression of Nectin 4 with RNA-sequencing and immuno-histochemistry.